# The Crucial Role of NLRP3 Inflammasome in Viral Infection-Associated Fibrosing Interstitial Lung Diseases

**DOI:** 10.3390/ijms221910447

**Published:** 2021-09-28

**Authors:** Wiwin Is Effendi, Tatsuya Nagano

**Affiliations:** 1Department of Pulmonology and Respiratory Medicine, Faculty of Medicine, Universitas Airlangga, Surabaya 60132, Indonesia; 2Division of Pulmonology and Respiratory Medicine Universitas Airlangga Teaching Hospital, Surabaya 60115, Indonesia; 3Division of Respiratory Medicine, Department of Internal Medicine, Kobe University Graduate School of Medicine, 7-5-1 Kusunoki-cho, Chuo-ku, Kobe 650-0017, Japan; tnagano@med.kobe-u.ac.jp

**Keywords:** NLRP3 inflammasome, IL-1β and IL-18, viral infection, inflammation, chronic respiratory diseases, idiopathic pulmonary fibrosis

## Abstract

Idiopathic pulmonary fibrosis (IPF), one of the most common fibrosing interstitial lung diseases (ILD), is a chronic-age-related respiratory disease that rises from repeated micro-injury of the alveolar epithelium. Environmental influences, intrinsic factors, genetic and epigenetic risk factors that lead to chronic inflammation might be implicated in the development of IPF. The exact triggers that initiate the fibrotic response in IPF remain enigmatic, but there is now increasing evidence supporting the role of chronic exposure of viral infection. During viral infection, activation of the NLRP3 inflammasome by integrating multiple cellular and molecular signaling implicates robust inflammation, fibroblast proliferation, activation of myofibroblast, matrix deposition, and aberrant epithelial-mesenchymal function. Overall, the crosstalk of the NLRP3 inflammasome and viruses can activate immune responses and inflammasome-associated molecules in the development, progression, and exacerbation of IPF.

## 1. Introduction

IPF is a severe, permanent, and chronic respiratory disease that causes lung parenchyma scars and stiffness, making it gradually worse over time and more complicated to breathe. IPF is as a critical public health problem with an estimated conservative incidence range of 3–9 cases per 100,000 per year for Europe and North America, and lower in East Asia and South America (less than 4 cases per 100,000) [1]. Based on a recent study, IPF incidence and prevalence are variable worldwide, about to be in the range of 0.09–1.30 and 0.33–4.51 per 10,000 persons, respectively; therefore, in all but South Korea, IPF would be classified as a rare disease according to national guidelines [2].

Despite less knowledge about the defined etiology and precise mechanisms, considerable advances in understanding IPF pathogenesis have been achieved in recent years. IPF is characterized by progressive extracellular matrix (ECM)-producing fibroblasts, fibroblast-myofibroblast transition, and extensive deposition of ECM by transforming growth factor-β1 (TGF-β1)-induced myofibroblasts [3,4]. Although IPF has been redefined due to repeated micro-injury of the alveolar epithelium and aberrant epithelial-mesenchymal transition (EMT) crosstalk in a genetically susceptible aging individual, chronic inflammation still plays an important role [5,6]. Inflammation also has contributions to IPF initiation and progression. Intrinsic factors, occupation, environmental influences, including exposure to microbes, particles, irritants, pollutants, allergens, and toxic molecules, genetic and epigenetic risk factors may lead to chronic inflammation that in turn leass to the development of IPF [7,8,9,10].

Innate and adaptive inflammation appears to be a prominent feature, markedly increased in the IPF group with rapid disease progression [11]. Along with previous studies, innate and adaptive immune and inflammatory cells produced heterogeneous contributions in remodeling and fibrosis processes [12]. Inflammation induced by viral infections and co-expression of virus-related proteins were firmly associated with IPF [13,14].

Upon infection with a respiratory virus, innate immune response signaling cascades start with the recognition of specific viral components, or pathogen-associated molecular patterns (PAMPs) and endogenous danger molecules that are generated and exposed from damaged or dying cells, damage-associated molecular patterns (DAMPs), by germline-encoded receptors called pattern recognition receptors (PRRs) [15,16]. PRRs are classified into four significant subfamilies: toll-like receptors (TLRs), nucleotide-binding oligomerization domain (NOD), leucine-rich repeat (LRR)-containing receptors (NLRs), retinoic acid-inducible gene-1 (RIG-1)-like receptors (RLRs), and C-type lectin receptors (CLRs) [17]. TLRs and CLRs control the extracellular environment and endosomal compartments, while RLRs and NLRs recognize microbial patterns or danger signals within the cellular cytoplasm [18].

In 2002, Martinon et al. discovered the inflammasome, a new platform for protein-containing PRRs [19]. Among various inflammasome complexes, the NLRP3 inflammasome has been under intensive investigation and linked with various human autoinflammatory and autoimmune diseases [20]. Viral infection-triggered NLRP3 inflammasomes play vital roles in the development of organ fibrosis [21,22,23]. Age-dependent increases in interleukin-1β (IL-1β) and IL-18 and alveolar macrophage mitochondrial reactive oxygen species (ROS) production in the development of experimental fibrosis was associated with NLRP3 inflammasome activation [24].

This brief review will focus on the recent research advances of NLRP3 inflammasome activation in response to viral infection throughout the development and progression of IPF. Identifying the mechanisms underlying the virus-induced NLRP3 inflammasome activation is essential to develop novel therapeutic strategies for lung fibrosis.

## 2. NLRP3 Inflammasome

The inflammasome is a multiprotein complex activated in response to microbial invasions or danger signals [25]. In sensing viral infection, inflammasomes will activate the innate immune response to regulate unwanted inflammasome activation and overt inflammation [26]. The classical inflammasome mainly activates caspase-1, whereas the non-classical inflammasome pathway is a caspase-1-independent inflammasome [27]. Indeed, inflammasomes regulate the secretion of pro-inflammatory cytokines, activate caspase-1, and the induction of pyroptosis as a mediator in the innate immune response [28]. When persistent inflammation occurs, activation of inflammasome complexes is initiated, leading to further activation of caspase-1, proinflammatory cytokines, and pyroptosis induction [29].

The complete structure of inflammasome is a sensor (NLRs, absent in melanoma 2 (AIM2)-like receptors (ALRs), and pyrin) that recognize a ligand, an adaptor (apoptosis-associated speck-like protein containing a caspase activation and recruitment domain (ASC)) and a zymogen pro-caspase-1 [30,31]. NLRs are cytosolic receptors widely identified in non-vertebrates and vertebrates, where humans express 22 NLR genes and at least 34 in mice [32,33]. NLRs are divided based on their specific N-terminal domain: NLRAs that have an acidic activation domain, NLRBs that possess a baculovirus inhibitor of apoptosis repeat (BIR)-like domain, NLRCs that feature a caspase activation and recruitment domain (CARD) or a Death domain (DD), and the NLRP subfamily that contain a pyrin domain (PYD) [34].

The NLRP subfamily comprises C-terminus (LRR), a central nucleotide-binding and oligomerization terminus (NOD/NACHT), and N-terminus (Pyrin domain or PYD) [35]. Several NLR and non-NLR families that can form multiprotein complexes inflammasome have been described, including NLRP3, NLRP1, AIM2, NLRC4, and pyrin. [36]. The NLRP3 inflammasome is assembled upon ligand recognition, the PYD of NLR sensor associates with the PYD of ASC, then recruits and activates caspase-1, leading to the proteolytic cleavage and secretion of the pro-inflammatory cytokines, IL-1β and IL-18, as well as to gasdermin D (GSDMD)-mediated pyroptotic cell death [37]. The NLRP3 inflammasomes are widely expressed in the cytoplasm of various cells, including fibroblast [38,39]. The structure of NLRP3 is depicted in Figure 1.

## 3. The Activation of NLRP3 Inflammasome

In general, fully activating the NLRP3 inflammasome (canonical activation) requires two steps: a priming signal (signal 1) and an activation step (signal 2). The binding of viral membrane components to PRRs, such as TLRs, NLRs, RLRs, or cytokine receptors initiates priming signals then activating the transcription factor nucleus factor-κβ (NF-κβ). NF-κβ upregulates the expression and translocation of NLRP3, pro-IL-1β, and -IL-18 from the nucleus to the cytosol, which remains inactive until stimulated by the second signal [40,41]. These priming signals also regulate post-translational modifications of inflammasome components, such as NLRP3 deubiquitination, ubiquitination, and ASC phosphorylation, which is needed for further activation of inflammasome complex assembly [42].

In case of deficient signaling molecules of the NF-κβ pathway, both IL-1R–associated kinase (IRAK)1 and IRAK4 bypass priming and directly link to TLR for rapid activation of NLRP3 through the MyD88 pathway [43,44]. The priming signals regulate NLRP3 inflammasome activation via both transcription-dependent and -independent pathways [45].

An additional stimulus (signal 2), induced by a wide range of stimuli following the priming step, is usually required to promote NLRP3 inflammasome assembly, leading to the oligomerization and activation of the inflammasome complex. The NLRP3 inflammasome complex triggers the cleavage of pro-caspase-1 into caspase-1, and subsequent maturation of pro-inflammatory cytokines IL-1β and IL-18, and pyroptosis induction [41]. Several molecular and cellular events have been proposed to trigger NLRP3 inflammasome activation, including K^+^ efflux, Ca^2+^ signaling, reactive organ species (ROS), mitochondrial dysfunction, ATP release, and lysosomal damage [35].

Apart from canonical activation, the NLRP3 inflammasome is switched on via non-canonical and alternative pathways. Unlike canonical NLRP3 inflammasome activation, non-canonical activation is critical for defense against intracellular Gram-negative bacteria [46,47] and requires a receptor, such as caspases 4/5 in humans [48,49] and caspase-11 in mice [50,51], rather than caspase-1. These caspases initiate non-canonical inflammasome activation through the direct recognition of intracellular LPS independently of TLR4 [46].

In contrast to classical NLRP3 inflammasome, human monocytes engage an alternative pathway that enables IL-1β secretion without K+ efflux, pyroptosome formation, and pyroptosis [52,53]. This signaling pathway involved TLR4–TRIF–RIPK1–FADD–CASP8 axis is limited to an alternative inflammasome and has no role in classical NLRP3 inflammasome activation [53].

## 4. Viral Infection Triggered The Activation of NLRP3 Inflammasome

NLRP3 inflammasome sensors are activated in response to both DNA and RNA viruses. Indeed, the NLRP3 inflammasome is essential in defending against viral infections (reviewed detailed in [54]). However, in the steady-state, inflammasome assembly is tightly regulated at a low level to prevent an aberrant pro-inflammatory response and cell death [55]. Viral infection activates the NLRP3 inflammasome via viral proteins, endoplasmic reticulum (ER) stress, mitochondria dysfunction, ROS production, protein aggregates, and aberrant ion concentrations [34]. In addition, both DNA and RNA viruses also activated the NLRP3 inflammasome through cathepsin B release [56]. The diversity of its activators makes the possibility of direct interaction between NLRP3 inflammasome with each activator unlikely [30].

During viral infection, DNA or RNA viruses that serves as PAMPs are recognized by a sensor of TLR or NLR results in the transcription and expression of the NLRP3 inflammasome, pro-IL-1β, and pro-IL-18 via NF-κβ upregulation (signal 1 or priming). In the context of signal 2 (activation), the three models that have been identified as DAMPs will activate NLRP3 inflammasome fully, including 1) ion channel model; 2) lysosomal rupture model; 3) mitochondria and ROS model. However, the precise mechanisms of virus-induced NLRP3 inflammasome activation are still not fully understood (Figure 2).

### 4.1. Viral DNA-Induced NLRP3 Inflammasome Activation

The NLRP3 identifies enveloped and non-enveloped DNA viruses, therefore it is a common pathway for viral detection [57]. Studies on non-enveloped viruses, such as adenovirus have shown different results, in that NLRP3 inflammasome activation was both NLRP3-dependent and NLRP3-independent, in which IL-1α, b3 integrins, cathepsin B- and ROS-dependent played a crucial role [58]. In addition, activation of inflammasome and transcription of IL-1β by modified vaccinia virus Ankara (MVA) infection is dependent on the crosstalk between TLR2-MyD88 and the NLRP3 [58].

Upon HSV-1 infection, cyclic GMP-AMP synthase (cGAS)-stimulator of interferon genes (STING)-NLRP3 signaling is required for the NLRP3 inflammasome activation and IL-1β secretion [59] in the presence of cytosolic DNA not RNA [60]. Previously, another study demonstrated HSV-1 infection induces the activation of the NLRP3 inflammasomes early during in vitro infection of fibroblasts [61].

### 4.2. Viral RNA-Induced NLRP3 Inflammasome Activation

Numerous studies demonstrated that RNA viruses could activate the NLRP3 inflammasome. Double-stranded RNA (dsRNA), such as influenza virus, induced IL-1β; release and caspase-1 activation via both NLRP3-dependent and NLRP3-independent mechanisms [58]. Influenza virus is the most common viral activator, and components of specific viruses can activate the NLRP3 inflammasome, including influenza virus proton-specific ion channel M2 protein, nonstructural protein PB1-F2, and vRNA [33]. Recent studies showed the roles of Z-DNA binding protein-1 (ZBP1) and IFN regulatory factor 1 (IRF1) transcription factor to promote NLRP3 inflammasome activation after influenza virus infection [62].

On a molecular level, viral RNA triggers NLRP3 inflammasome activation through RNA-modulating proteins, including DHX33, RNase L, RIG-1, and ROS production [62]. DDX19A, a member of the DEAD/H-box protein family, bridged viral RNA and NLRP3 to activate the NLRP3 inflammasome [63]. A study revealed that mitochondrial mitofusin protein 2 (MFN2), involved in cytoprotection, was required for NLRP3 inflammasome activation in response to RNA viruses [64].

Overall, various RNA viruses can activate the NLRP3 inflammasomes via ROS-dependent pathway, rupture lysosomes, the Golgi network, cathepsin B-dependent process, MFN2, Ca^2+^ model, and K^+^ model involving pannexin-1 and P2X purinoceptor 7 (P2X7) channels [65]. Furthermore, mechanisms of virus-mediated NLRP3 activation through viral RNA (single-stranded RNA (ssRNA) or dsRNA) depending on the type of virus and by viroporin proteins, which induce ion efflux from intracellular storage into the cytosol [58].

### 4.3. Viroporin-Induced NLRP3 Inflammasome Activation

Among the viral proteins, viroporins are present in both DNA and RNA viruses involved in different stages of the virus infection cycle and can induce NLRP3 inflammasome activity [66]. Viroporins are small, non-glycosylated, highly hydrophobic proteins, which interact with cell membranes and increases their permeability to ions and other small compounds [67].

Activation of NLRP3 inflammasomes-associated viroporin activity can be clustered into three main groups, including: (1) viroporins that pump protons and dissipate the proton gradient across trans-Golgi network, e.g., M2, SH, VP, P7, and E viroporins, (2) viroporins that manipulate Ca^2+^ homeostasis, stimulating Ca^2+^ flux from intracellular storages to the cytosol providing the second signal for NLRP3 activation and IL-1β production, such as 2B and P7 viroporins, (3) viroporins that increase mitochondrial stress and affects ROS production, such as 3a and M2 viroporins [66].

### 4.4. Ionic Flux Disturbance-Induced NLRP3 Inflammasome Activation

Inflammasomes can sense falls in cytosolic ion dysregulation. Ionic flux events, including K^+^ efflux, Ca^2+^ mobilization, Cl¯ efflux, and Na^+^ influx are implicated in activating the NLRP3 inflammasome. A decrease in intracellular K^+^ concentration was identified as the common trigger of NLRP3 inflammasome activation in response to ATP, nigericin, particulate molecules, crystalline, small chemical compounds, such as GB111-NH2, imiquimod, and CL097, NLRP3-activating mutations, and signals from the alternative inflammasome pathway [45]. Lower and higher concentrations of cytosolic K^+^ will induce and block the activation of the NLRP3 inflammasome via a mechanism that is still not well understood [65].

Replicating RNA viruses with a cytopathogenic effect can induce NLRP3 inflammasome activation via K^+^ efflux [68]. Activation of the NLRP3 inflammasome via some Picornaviridae viruses, such as EMCV and EV71, was independent of mitochondrial ROS and lysosomal cathepsin B, but dependent on ion efflux [69].

Mobilization of Ca^2+^ occurs by opening plasma membrane channels or the release of ER-linked intracellular Ca^2+^ stores to enable the flux of Ca^2+^ to the cytosol [37]. Interestingly, Negash et al. showed that HCV infection could induce NLRP3 inflammasome activation via K^+^ efflux in Kupffer cells [23] and via Ca^2+^ mobilization linked with phospholipase-C activation [70]. In addition, Ca^2+^ mobilization-induced signaling depends on another molecule pathway to integrate sufficient signals for NLRP3 inflammasome activation; hence the cross-talk between these ion efflux pathway components is complex, regulated in a context-dependent manner and remains to be fully elucidated [71]

Low extracellular Cl¯ levels promoting ASC polymerization during NLRP3 inflammasome formation, enhance ATP-induced IL-1β secretion [37]. A study showed that chloride intracellular channels (CLIC)-dependent chloride efflux has an essential role and acts downstream of the K^+^ efflux -mitochondrial ROS axis to promote NLRP3 inflammasome activation [72].

Regarding ion fluxes, K^+^ efflux is a distal upstream event while Cl^−^ efflux is a proximal upstream event for NLRP3 inflammasome activation and acts downstream of mitochondrial damage and Ca^2+^ release from the endoplasmic reticulum, promoting mitochondrial damage and subsequent NLRP3 inflammasome activation [73].

### 4.5. Intracellular ROS and Mitochondria Dysfunction-Induced NLRP3 Inflammasome Activation

Mitochondria are suitable targets for viruses. Viral infection induces mitochondrial dysfunction via post-translational modification, regulation of expression, disturbs mitochondrial dynamic proteins and modifies the physiological environment within cells [74]. During airway and parenchymal inflammation, mitochondria can release mitochondrial reactive oxidative species (mtROS), leading to the activation of NLRP3 inflammasome [75]. The precise role of mtROS in NLRP3 inflammasome activation remains controversial because ROS may be required only during the transcriptional priming step rather than for post-translational NLRP3 activation itself [76].

Ad virus, Flaviviridae family, influenza virus, and myxoma virus activate the NLRP3 inflammasome via mtROS and cathepsin B induction [58]. Dengue virus and other RNA viruses mediate mitochondria aberrations and ROS production leads to NLRP3 inflammasome activation through the RIP1-RIP3-DRP1 pathway [54].

Meanwhile, mitochondrial-associated molecules, such as mitochondrial antiviral-signaling protein (MAVS), MFN2, and cardiolipin, have also been implicated in NLRP3 activation. Therefore, RNA viruses but not non-viral stimuli, such as ATP or nigericin can activate the NLRP3 inflammasome via MAVS proteins and MFN2 [35,45]. Murine norovirus (MNV) leads to GSDSD dependent pyroptosis resulting in NLRP3 activation exhibited elevated MAVS-mediated IL-1β secretion. Also, MFN2 is required for NLRP3 activation after infection with RNA viruses, such as influenza, measles, or encephalomyocarditis virus (EMCV) [65].

Metabolic activity and redox state are all intricately linked to each other and connect mitochondrial network dynamics during infection and NLRP3 inflammasome activity [77]. All this evidence supports mitochondria as central regulators of NLRP3 inflammasome activation induced by ER stress, virus infections and the NLRP3 activators accompanying mitochondrial dysfunction to promote the activation of NLRP3 inflammasome [78].

### 4.6. ER Stress-Induced NLRP3 Inflammasome Activation

The mechanism of ER stress that triggers the NLRP3 inflammasome function is unclear. Dysregulation of ER functions leads to the accumulation of misfolded- or unfolded-protein in the ER lumen, which triggers the unfolded protein response (UPR), repairing ER stress and dysfunction via ER-resident proteins inositol-requiring enzyme 1 (IRE1), protein kinase R (PKR)-like ER kinase (PERK), and activating transcription factor 6 (ATF6) [79]. Viral infection induces ER stress and UPR to promote cell survival by inhibiting apoptosis and maintains an environment favorable for the establishment of persistent infection [80]. Porcine reproductive and respiratory syndrome virus (PRRSV) utilizes the UPR machinery to potentiate the UPR, hijacks ATF4 for viral replication complexes, and promotes viral RNA replication [81].

Along with that, ER stress also triggers the NLRP3 inflammasome. The molecular mechanisms ER stress activates the NLRP3 inflammasome are still poorly understood. Kinase receptor-interacting protein 1 (RIP1) plays a critical role in ER stress-induced activation of the NLRP3 inflammasome through regulating mitochondria fission factor dynamin-related protein 1 (DRP1) and mtROS [82]. Previously, Bronner et al. demonstrated that ER stress-induced the inflammasome via NLRP3- and Caspase-2 required mitochondrial damage and mtROS production [83]. Furthermore, ER stress triggers the NLRP3 inflammasome through several proposed mechanisms, ER stress alone is sufficient for the induction of IL-1β production via the activation of the NF-κB and NLRP3 inflammasome pathways [84]; however, knockdown of PERK, IRE1α, and ATF6α did not affect ER stress-induced inflammasome activation [85].

Latent human herpesviruses (HHV) infection may impact telomere length by various mechanisms and activate UPR in cultured human fibroblasts by modifying ER stress proteins to favor viral persistence. Therefore, there appear to be several plausible mechanisms by which HHV could also enable or promote lung fibrosis via ER stress, telomere attrition, and cellular senescence in aging individuals [86]. In addition, RNA viruses alter ER function via the exploitation of the ER membrane, rapid accumulation of misfolded proteins, imbalance of calcium concentration by viroporin, and the sabotage or depletion of the ER membrane during virion release [87].

Overall, viral infection triggers ER stress- and NLRP3 inflammasome-induced inflammation signaling cascades via complex mechanisms that involve cell stressors, such as mitochondrial stress, cytokines, RIP1, and Angiotensin II [88].

### 4.7. Lysosomal Damage-Induced NLRP3 Inflammasome Activation

PAMPs and DAMPs of viral origin and NLRP3 activators, including nigericin, asbestos, silica, alum, and amyloid β, form aggregates that are phagocytosed resulting in the disruption of lysosomes and the consequent release of lysosomal components, especially cathepsin B [58]. Influenza viruses use rupture lysosomes and then activate NLRP3 inflammasomes via a cathepsin B-dependent processes; however, the function of these events remains an open question in NLRP3 activation [65].

## 5. Potential Mechanism of Virus Infection-Induced Damage That Associated with The NLRP3 Inflammasome

Respiratory virus infection has a crucial role in IPF pathogenesis; either as a cofactor in the development of IPF due to the presence of persistent viruses before incidence, or through the induction of inflammation via ER stress and apoptosis in epithelial cells which leads to acute exacerbation of IPF (AE-IPF) [89]. Human torque teno viruses (TTV) infections are associated with the severity of various diseases, such as hepatitis, IPF, and autoimmune disorders [90]. Some studies found the pathogenetic link between pulmonary fibrosis and Hepatitis C virus (HCV) infection [91,92]. More recently, Samir et al., by using a high-resolution CT scan (HRCT), detected a fibrotic pattern of ILD in patients with high HCV viremia and showed a restrictive diffusing capacity of carbon monoxide (DLCO) decline pattern [93].

It has been assumed that persistent virus infection facilitates exogenous injury for the development of pulmonary fibrosis in susceptible individuals. In contrast, the direct impact of viral infection is insufficient to trigger pulmonary fibrosis. However, Zhou et al. instilled high dose adenoviruses (Ad) vectors into mouse lungs resulting in inflammatory responses, lung injury, and pulmonary fibrosis in a dose-dependent manner [94].

The HHV family (herpes simplex virus type 1 (HSV-1), Epstein–Barr virus (EBV), cytomegalovirus (CMV), and HHV-7 and -8) has received the most significant attention as either an etiologic or exacerbating agent because these viruses can establish life-long latency with the potential to reactivate disease in older individuals [95]. A Recent study demonstrated EBV and HHV-8 DNA in the lung tissue of IPF specimens [96].

Once infected, HHV becomes latent within the nucleus and persists in episomal form, periodically reactivating as a lytic virus (a lytic form of cell death) [97]. Li and his colleagues showed that latent CMV infection aggravated pulmonary fibrosis, possibly through the activation of TGF-β1 [98]. Furthermore, HSV-1 induced transcription of molecular pathways that promote fibrotic in IPF [99]. Supporting the previous study, Karaba et al. found HSV-1 activates canonical and non-canonical NLRP3 inflammasome pathways to lead to the release of IL-1β [100]. In a bleomycin-induced mouse model, scutellarin suppressed excessive EMT and inflammation via NF-κβ/NLRP3 pathway [101]. Briefly, chronic infection of HHV significantly increases the risk of developing IPF but not for AE-IPF [102].

The phases model of abnormal wound healing in lung fibrosis includes (1) micro-injury; (2) chronic inflammation; (3) repair; and (4) fibrosis [103]. The expression and activity of the NLRP3 inflammasome from different cells in those phases are various. After the injury and viral infection, macrophage NLRP3 inflammasomes secrete pro-inflammatory cytokines (IL-1β and IL-18), followed by fibroblast proliferation and ECM synthesis induced by the myofibroblast NLRP3 inflammasome and finally, the epithelial cell NLRP3 inflammasome contributes to EMT, that contributes to fibrogenesis via interaction with TGF-β1 [104]. Overall, the NLRP3 inflammasome-viruses crosstalk can activate immune responses and inflammasome-associated molecules in lung fibrogenesis.

Molecular mechanisms of lung injury resulting in inflammation and fibrosis are not entirely explained; however, viral infection-associated NLRP3 inflammasomes might have a role in driving lung fibrosis. There are some pieces of evidence to suggest that the viral infection-associated NLRP3 inflammasome can drive fibrosis. The NLRP3 inflammasome was required to regulate the healing of damaged lung tissues after influenza A virus (IAV) infection [105]. Another study revealed that activation of the NLRP3 inflammasome pathways may contribute to pulmonary fibrosis caused by latent murine cytomegalovirus (MCMV) infection [22].

In the COVID-19 pandemic, some evidence supports the involvement of inflammasomes in SARS-CoV-2-derived interstitial fibrosis and diffuse alveolar damage [106]. Moreover, SARS-CoV-2 might directly activate the NLRP3 inflammasome, resulting in the activation of macrophages, neutrophil infiltration, reduced apoptosis and excessive cytokine production which can lead to cytokine storms and fibrosis [107]. Overall, age and sex difference-related NLRP3 inflammasome activation in SARS-CoV-2 infections increased lung inflammation and fibrosis [108,109].

### 5.1. Inflammasome-Derived Inflammatory Cytokines

Following full activation of the NLRP3 inflammasome, pro-IL-1β and pro-IL-18 are completely equipped into mature biologically active proteins and critically required in wound healing and fibrosis. IL-1β secreted by alveolar macrophage promotes the recruitment of neutrophils and lymphocytes, and promotes collagen synthesis, leading to inflammation at the injury site. IL-18 induces EMT and lung fibroblast senescence by downregulating the anti-senescence protein Klotho in pulmonary fibrosis [110].

In the presence of HHV, there was increased expression of TGF-β1, VEGF, chemokines, particularly monocyte chemotactic protein (MCP)1/CCL2, and T-helper (Th)2-chemokines, such as macrophage inhibitory protein (MIP)1α/CCL3, CCL4, CCL5, and CCL12 [86]. Virus infection-induced NLRP3 inflammasomes generate hyper-immune responses and inflammation by various interleukins and cytokines and infiltration of other immune cells (mainly fibroblasts, macrophages, and alveolar epithelial cells (AECs)) in the formation and development of pulmonary fibrosis by regulating inflammation, immune response, autophagy, senescence, EMT, potentially leading to pulmonary fibrosis [110,111]. Crosstalk between the presence of lytic or latent herpes viruses and modulation of cytokines regulate a cell signaling environment conducive to the development of lung fibrosis from prior or subsequent lung injury [86].

Viral infection-associated NLRP3 inflammasome-dependent inflammation mediates the progression of lung fibrosis by escalating the inflammatory response in immune cells and the cross-talk between immune cells and growth factors. A study found that galectin-3 (gal-3) plays a crucial role in H5N1-induced inflammation by promoting host inflammatory responses, NLRP3 inflammasome assembly, and enhancing IL-1β production [112]. Therefore, targeting gal-3 inhibition was associated with reductions in plasma biomarkers centrally relevant to IPF pathobiology [113].

However, viral infection can induce IL-1β–mediated inflammation through NLRP3-independent mechanisms. Vaccinia virus (VACV) infection induces NLRP3-independent maturation of caspase-1 [114]. Next, Ichinohe et al. demonstrated that IL-1β secretion in resident hematopoietic cells due to IAV infection was NLRP3-independent [115]. A current study showed that in IAV infection, inflammasome formation in respiratory epithelial cells was not dependent on NLRP3 but somewhat dependent on myxovirus resistance protein 1 (MxA) [116].

### 5.2. Pyroptosis and Necroptosis

IPF is a disease associated with age-related apoptosis. Apoptosis is an essential intracellular mechanism for maintaining homeostasis in multicellular organisms for removing unwanted and damaged cells; however, it is no longer solely synonymous with programmed cell death because of the identification of other forms of programmed death, such as necroptosis and pyroptosis [117]. Both necroptosis and pyroptosis will release inflammatory intracellular molecules, leading to inflammation.

Damage, aberrant senescence, increased apoptosis of AECs but decreased apoptosis of fibroblasts, and dysfunctional repair are regarded as the core pathogenesis of IPF [110,118]. Based on a current study, pyroptosis and release of inflammasome components increased IL-1β secretion, and α-smooth muscle actin expression results in liver injury and liver fibrosis development [119]. Also, NLRP3 inflammasome-mediated pyroptosis provokes pressure overload-induced cardiac hypertrophy and fibrosis in mouse models [120].

During IAV infection, an innate immune sensor and the interferon-inducible ZBP1 can sense Z-RNA and trigger cell death through PANoptosis (pyroptosis, apoptosis, and necroptosis) lead to the formation of the ZBP1-NLRP3 inflammasome [71]. Moreover, IAV infection induces alveolar epithelial damage and infiltration of inflammatory cells in lung-related necroptosis, which may involve ZBP1 [121].

Age-related mechanisms that diminish the regenerative capacity of the aged lung lead to IPF [122]. Takezaki et al. revealed that Sftpa1 knock-in (Sftpa1-KI) mice increased necroptosis of AECs that was accelerated by influenza virus infection in IPF [123]. In SARS-CoV-2 infection, the NLRP3 inflammasome is over-activated in aged individuals through mitochondrial dysfunction leading to hyper inflammation response and subsequent increases in IL-1β [108]. Recently, a study found that age-dependent activation of NLRP3 inflammasome involved in the activation of TGF-β and deposition of ECM in mice model [124].

## 6. Conclusions

In senescence-related susceptible individuals exposed to endogenous (genetics and epigenetics) stress and exogenous (environment and infection) injury, virus-induced NLRP3 inflammasome activation may drive the inflammation response by activating caspase-1 and production of mature IL-1β and IL-18. Both latent virus and lytic forms can activate the NLRP3 inflammasome by integrating cellular signaling involving mitochondria ROS, lysosome rupture, ER stress, and ion flux. Furthermore, NLRP3 inflammasome-associated cytokine secretion implicates robust inflammation, fibroblast to myofibroblasts differentiation, synthesis of ECM, and EMT. In light of this, NLRP3 inflammasome-related molecules associated with the viral infection may contribute to the development, preservation of progression, and triggering exacerbation of IPF. Further studies on the activation of the NLRP3 inflammasome by the virus form, latent or lytic, are needed to elucidate inflammasome-derived fibrosis pathogenesis.

## Figures and Tables

**Figure 1 ijms-22-10447-f001:**
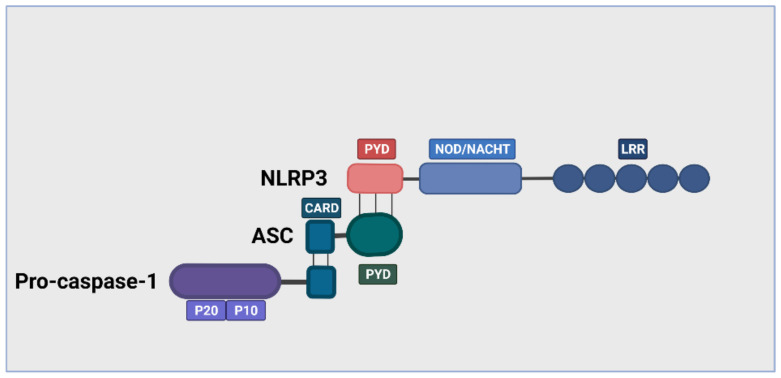
Structure of the complex NLRP3 inflammasome. The NLRP3 inflammasome contains an NLRP3 receptor, an ASC adaptor, and pro-cysteinyl aspartate specific proteinase-1 (pro-caspase-1). NLRP3 comprises of carboxyl (C) terminal (LRR), a central terminal, and an amino acid (N) terminal (PYD). The sensor C-terminal LRR recognizes PAMPs and DAMPs, the central terminal NACHT encodes by NLRP3 (CIAS1) gen, and the N-terminal CARD interacts with the adaptor protein through PYD-PYD. ASC recruits procaspase-1 through CARD domain to activate downstream signals.

**Figure 2 ijms-22-10447-f002:**
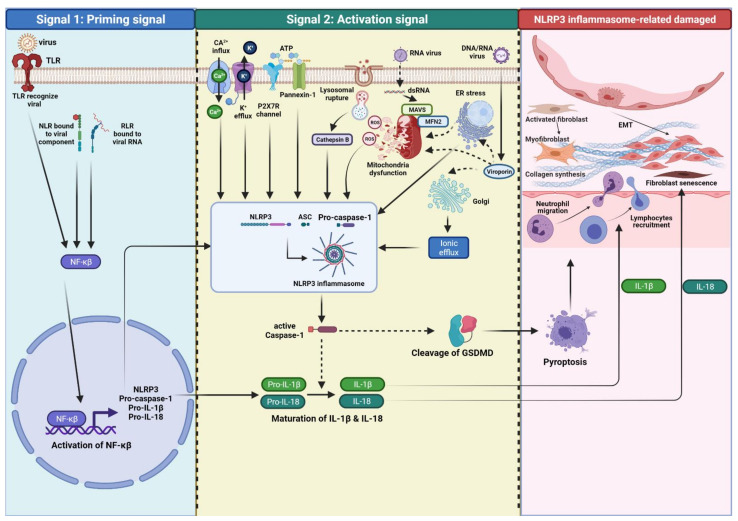
Overview mechanism of viral infection-induced NLRP3 inflammasome activation that drives fibrogenesis in IPF. NLRP3 inflammasome activation involves signal 1 for priming and signal 2 for complex protein assembly. Signal 1 is triggered by a DNA virus or RNA virus that is recognized as PAMPs via TLR, NLR, and RLR, leading to the upregulation and activation of the NLRP3 inflammasome, pro- caspase-1, pro-IL-1β, and pro-IL-18 via NF-κβ activation. An additional stimulus (signal 2) is induced by multiple molecular or cellular events (DAMPs), such as lysosomal damage, mitochondrial ROS, trans-Golgi network, ER stress, K^+^ efflux and Ca^2+^ influx, K^+^ model involving pannexin-1 and P2X7 channels. These signals promote assembly of the NLRP3 inflammasome to ASC, which then recruits pro-caspase-1 leads to the oligomerization and activation of the NLRP3 inflammasome complex. NLRP3 inflammasome activation leads to the auto-cleavage of pro-caspase-1. Activation of caspase-1 induces pyroptosis of infected cells via cleavage of GSDMD and proteolytic process of pro-IL-1β, pro-IL-18. Pyroptosis, IL-1β, and IL-18 are NLRP3 inflammasome-associated cellular event cytokines that implicate robust inflammation, fibroblast to myofibroblasts differentiation, synthesis of ECM, and EMT.

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
