# Peer review of "The Crucial Role of NLRP3 Inflammasome in Viral Infection-Associated Fibrosing Interstitial Lung Diseases"

_ijms, 2021, doi:10.3390/ijms221910447_

Round 1

Reviewer 1 Report

The work is clear and well described. Each paragraph is accurately discussed. The bibliography is relevant and the images are self-explanatory.

After reading the paragraphs, the arguments seem presented clearly. The work is well structured, and the text is easy to read. References to related and previous work are appropriate and adequate. The topic is original, and the functional investigation of NLRP3 inflammasome-viruses crosstalk is very interesting. Molecular mechanisms underlying viral infection-induced NLRP3 inflammasome activation that drives fibrogenesis are comprehensively discussed. Moreover, this manuscript opens new insight about the role of viral infection and NLRP3 inflammasome in driving lung fibrosis.

Author Response

Review report (reviewer 1)

The work is clear and well described. Each paragraph is accurately discussed. The bibliography is relevant and the images are self-explanatory.

After reading the paragraphs, the arguments seem presented clearly. The work is well structured, and the text is easy to read. References to related and previous work are appropriate and adequate. The topic is original, and the functional investigation of NLRP3 inflammasome-viruses crosstalk is very interesting. Molecular mechanisms underlying viral infection-induced NLRP3 inflammasome activation that drives fibrogenesis are comprehensively discussed. Moreover, this manuscript opens new insight about the role of viral infection and NLRP3 inflammasome in driving lung fibrosis.

Author`s response

Thank you for your appreciation. I hope my work will give more information regarding viral infection-induced NLRP3 inflammasome activation that drives lung fibrosis.

Reviewer 2 Report

  1. Idiopathic pulmonary fibrosis (IPF) is a disease of unknown origin -- and experimental evidence suggests that viral infections MAY play a role, either as agents that predispose the lung to fibrosis or exacerbate existing fibrosis.
    I suggest excluding the word "Idiopathic" from the title or change the title to exclude misunderstanding
  2. Also, it will be good to review biomarkers associated with  NLRP3-dependent and NLRP3-independent inflammation 

Author Response

Review report (reviewer 2)

  • Idiopathic pulmonary fibrosis (IPF) is a disease of unknown origin -- and experimental evidence suggests that viral infections MAY play a role, either as agents that predispose the lung to fibrosis or exacerbate existing fibrosis.
    I suggest excluding the word "Idiopathic" from the title or change the title to exclude misunderstanding
  • Also, it will be good to review biomarkers associated with  NLRP3-dependent and NLRP3-independent inflammation 

Author`s response

Thank you for your suggestion

  • I modified the title with the new one, “The Crucial Role of NLRP3 Inflammasome in Viral Infection-Associated Fibrosing Interstitial Lung Diseases”. Idiopathic pulmonary fibrosis (IPF) is included in ILD that is characterized by chronic progressive fibrosis.
  • I discussed NLRP3-dependent and NLRP3-independent mechanisms that drive inflammation and fibrosis in virus infection in lines 399-412.

Reviewer 3 Report

The manuscript "The Crucial Role of NLRP3 Inflammasome in Viral Infection-
Associated Idiopathic Pulmonary Fibrosis" is and interesting review article. The manuscript is well written and the reading pleasant: it will be interesting for the readers of the International Journal of Molecular Sciences. Few comments:

  • Due to the rising evidence of NLRP3 involvment in SARS-CoV-2 related disease (COVID-19) and its evolution to pulmonary fibrosis, I would suggest the authors to develop a whole paragraph related to it. (https://www.ncbi.nlm.nih.gov/pmc/articles/PMC7332883/; https://www.nature.com/articles/s41577-021-00588-x )
  • A recently, developed animal model of fibrosis showed inflammasome activation (https://www.mdpi.com/1422-0067/22/16/8833). This is in agreement with lines 65-68. The manuscript could be improved by a breif discussion of age-dependent mechanisms of NLRP3 activation.

Author Response

Review report (reviewer 3)

The manuscript "The Crucial Role of NLRP3 Inflammasome in Viral Infection-
Associated Idiopathic Pulmonary Fibrosis" is an interesting review article. The manuscript is well written and the reading pleasant: it will be interesting for the readers of the International Journal of Molecular Sciences. Few comments:

  • Due to the rising evidence of NLRP3 involvment in SARS-CoV-2 related disease (COVID-19) and its evolution to pulmonary fibrosis, I would suggest the authors to develop a whole paragraph related to it. (https://www.ncbi.nlm.nih.gov/pmc/articles/PMC7332883/; https://www.nature.com/articles/s41577-021-00588-x)
  • A recently, developed animal model of fibrosis showed inflammasome activation (https://www.mdpi.com/1422-0067/22/16/8833). This is in agreement with lines 65-68. The manuscript could be improved by a brief discussion of age-dependent mechanisms of NLRP3 activation.

Author`s response

Thank you for your correction and good suggestion.

  • I add some recent information about NLRP3 inflammasome activation-associated lung inflammation and fibrosis in SARS-CoV-2 infection in line 374-389.
  • It is a good idea to discuss age-related NLRP3 inflammasome induce lung fibrosis. IPF is a disease associated with age-related apoptosis; therefore, I add a brief discussion of that idea in lines 432-439.